# The molecular basis of Abelson kinase regulation by its αI-helix

**Johannes Paladini†, Annalena Maier†‡, Judith Maria Habazettl, Ines Hertel, Rajesh Sonti§, Stephan Grzesiek***

Structural Biology and Biophysics, Biozentrum, University of Basel, Basel, Switzerland

**\*For correspondence:**
stephan.grzesiek@unibas.ch

†These authors contributed equally to this work

**Present address:** ‡Tissue Engineering + Biofabrication Laboratory, Department of Health Sciences & Technology, ETH Zurich, Zurich, Switzerland; §Department of Pharmaceutical Analysis, National Institute of Pharmaceutical Education and Research (NIPER), Hyderabad, India

**Competing interest:** The authors declare that no competing interests exist.

**Abstract** Abelson tyrosine kinase (Abl) is regulated by the arrangement of its regulatory core, consisting sequentially of the SH3, SH2, and kinase (KD) domains, where an assembled or disassembled core corresponds to low or high kinase activity, respectively. It was recently established that binding of type II ATP site inhibitors, such as imatinib, generates a force from the KD N-lobe onto the SH3 domain and in consequence disassembles the core. Here, we demonstrate that the C-terminal αI-helix exerts an additional force toward the SH2 domain, which correlates both with kinase activity and type II inhibitor-induced disassembly. The αI-helix mutation E528K, which is responsible for the ABL1 malformation syndrome, strongly activates Abl by breaking a salt bridge with the KD C-lobe and thereby increasing the force onto the SH2 domain. In contrast, the allosteric inhibitor asciminib strongly reduces Abl's activity by fixating the αI-helix and reducing the force onto the SH2 domain. These observations are explained by a simple mechanical model of Abl activation involving forces from the KD N-lobe and the αI-helix onto the KD/SH2SH3 interface.

## eLife assessment

This manuscript describes an **important** NMR investigation of allosteric interactions within Abl kinase. The authors identify helix I as a major element that couples the Abl active site with the myristate-binding pocket. The **convincing** findings have implications for understanding Abl kinase activation and how to target Abl kinase in diseases.

## Introduction

Abelson tyrosine kinase (Abl) is crucial for many cellular processes including proliferation, division, survival, DNA repair and migration (*Van Etten, 1999*; *Pendergast, 2002*). Under non-pathological conditions, Abl is tightly regulated with very low intrinsic activity in unstimulated cells (*Hantschel, 2012*). However, the oncogenic t(9;22)(q34;q11) chromosome translocation (Philadelphia chromosome) leads to the expression of the highly active fusion protein Bcr-Abl and subsequently to chronic myeloid leukemia (CML; *Rowley, 1973*; *Deininger et al., 2000*; *Braun et al., 2020*). The orthosteric ATP site inhibitors imatinib (Gleevec), nilotinib (Tasigna), and dasatinib (Sprycel) constitute the front-line therapy against CML (*Hantschel et al., 2012*; *O'Hare et al., 2009*; *Shah et al., 2007*), but the emergence of drug-resistant point mutations in a fraction of patients has created the need for alternatives (*Hantschel et al., 2012*; *Eide et al., 2019*). In particular, the recently FDA-approved allosteric inhibitor asciminib (ABL001; *Wylie et al., 2017*), which targets the myristoyl binding pocket (STAMP, specifically targeting the ABL myristoyl pocket), shows high promise to overcome these resistances (*Réa et al., 2021*). This development now enables dual Bcr-Abl targeting and thereby therapy of patients bearing compound mutations (*Wylie et al., 2017*; *Eide et al., 2019*). However, the exact molecular mechanism of the interplay between allosteric and orthosteric inhibition is currently still unclear.

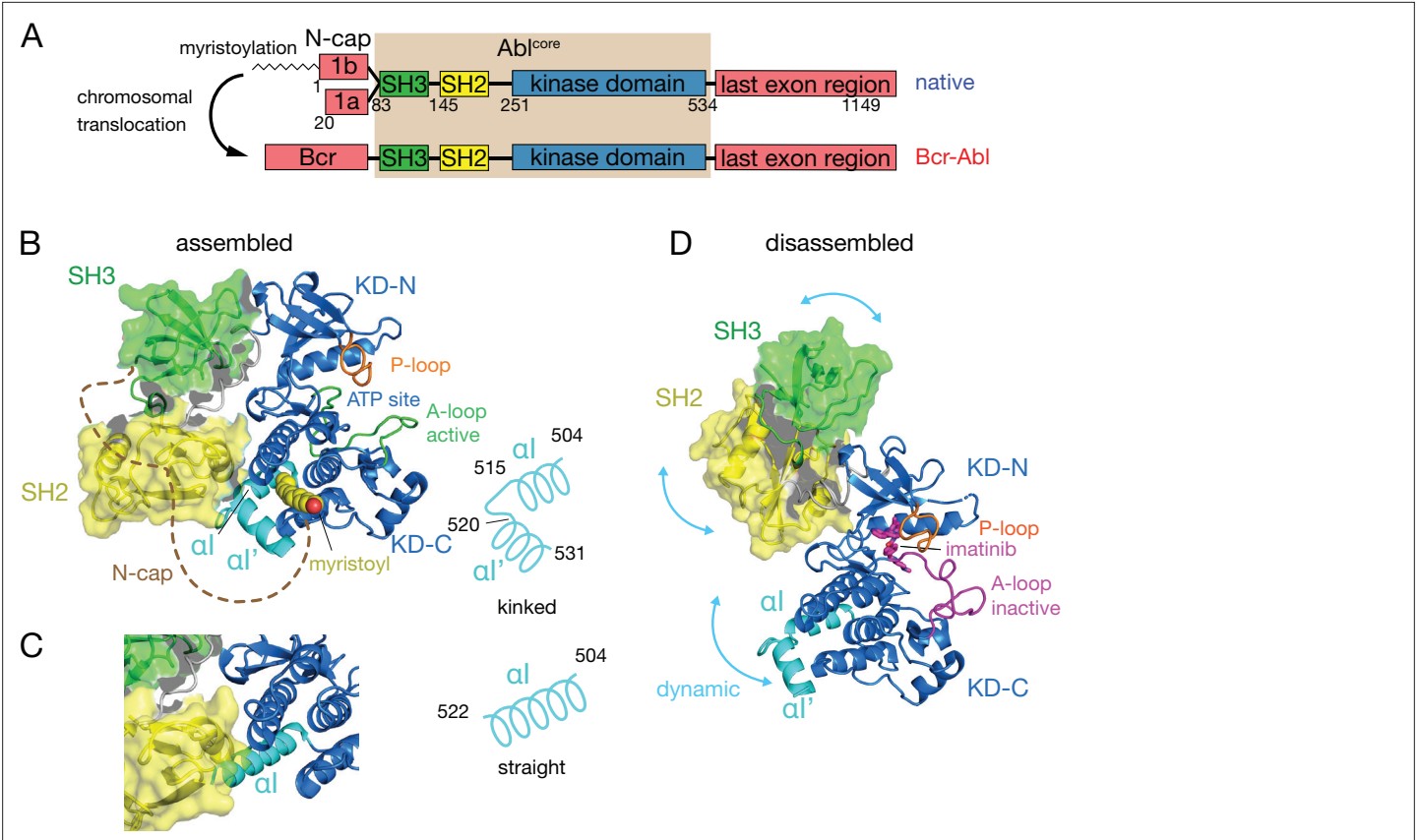

**Figure 1.** Domain organization of Abl and ligand-induced conformations of the Abl core. (**A**) Domain organization of Abl isoforms 1a,b and of the chronic myeloid leukemia (CML)-inducing fusion protein Bcr-Abl. (**B**) Crystal structure of the N-terminally myristoylated Abl 1b isoform (residues 2–531, PDB 2FO0). SH3 domain (green), SH2 domain (yellow), KD-SH2 linker (grey), kinase domain (KD) N- and C-lobes (KD-N and KD-C, respectively, blue), activation loop (A-loop, green), and αI-helix (cyan) are shown in cartoon representation. The myristoyl is shown as yellow spheres. The N-cap (aa 2–82), which is largely not observed in the crystal, is indicated as a dashed brown line. A schematic representation on the right indicates the position of the kink in the αI-αI′ helix. (**C**) Alignment of the crystal structure of a KD-only construct with unliganded myristoyl pocket (blue, PDB 1M52) with the assembled Abl core structure (PDB 2FO0). The SH3 and SH2 domains of the latter are displayed as in panel B. (**D**) Model of a single conformation of the dynamical imatinib-bound, disassembled Abl core as derived by NMR and SAXS (*Skora et al., 2013*). The coloring follows panel B with the exception of the activation loop (magenta). Imatinib is shown as magenta sticks. Blue arrows indicate relative motions of the SH3, SH2, and KD domains.

Under healthy conditions, Abl regulation is achieved by a set of interactions within its regulatory core, consisting sequentially of the SH3, SH2 and kinase (KD) domains, and the preceding ~60–80-residue-long N-terminal tail (N-cap) (*Nagar et al., 2003*; *Figure 1A*). The N-cap varies between splice variants 1a and 1b with Abl 1b being 19 residues longer and N-terminally myristoylated. Crystal structures of the autoinhibited Abl 1b core with the myristoylated N-cap (*Nagar et al., 2003*; *Hantschel et al., 2003*; *Nagar et al., 2006*) reveal a tight, almost spherical assembly (*Figure 1B*). This assembly appears stabilized by several interactions: (i) the docking of the SH3 domain to the proline-rich linker that connects SH2 domain and KD N-lobe; (ii) extensive contacts of SH3 domain and KD N-lobe as well as SH2 domain and KD C-lobe and (iii) the docking of the N-terminal myristoyl into a hydrophobic cleft at the bottom of the KD C-lobe. The assembly of the core impedes efficient substrate binding (*Nagar et al., 2003*), presumably by hindering hinge motions between the KD C- and N-lobes (*Sonti et al., 2018*), and reduces the kinase activity by 10- to 100-fold relative to the isolated kinase domain (*Hantschel, 2012*; *Sonti et al., 2018*). In contrast, the active state of Abl is thought to require the disassembly of the core to make the substrate binding site accessible and expose the protein-protein contact sites of the SH2 and SH3 domains.

The C-terminal αI-helix (residues 504–522, 1b numbering used throughout) adopts a straight conformation (*Figure 1C*) in crystal structures of the isolated Abl kinase domain with an empty myristoyl binding pocket (PDB 1M52; *Nagar et al., 2002*). In contrast, in the assembled core structures

(*Figure 1B*, PDB 2FO0; *Nagar et al., 2006*), the helix breaks into two parts αI (residues 504–515) and αI' (residues 520–531) connected by the αI–αI' loop, with the αI' part bending toward the myristoyl bound in the myristoyl pocket (*Figure 1B*). Notably, the straight αI-helix of the isolated Abl kinase domain would clash with the SH2 domain in the assembled core (*Figure 1C*). This has led to the notion that myristoyl binding induces a bend of the αI-helix, thereby stabilizing the assembled core and reducing activity (*Hantschel et al., 2003*; *Nagar et al., 2003*). Following this idea, allosteric inhibitors have been developed that target the myristoyl pocket (*Adrián et al., 2006*; *Jahnke et al., 2010*; *Zhang et al., 2010*), leading to the drug asciminib (*Wylie et al., 2017*). Importantly, not all myristoyl pocket binders act as allosteric inhibitors, but only those that also bend the αI-helix (*Jahnke et al., 2010*). The importance of this structural region for Abl's function is highlighted by several mutations in the αI'-helix, including E528K at its C-terminal end associated with the recently described ABL1 malformation syndrome leading to congenital heart disease, skeletal malformations, characteristic facies, and hearing impairment (*Wang et al., 2017*; *Blakes et al., 2021*). The disruption of interactions by these mutations has been hypothesized as the cause for a concomitant increase in kinase activity (*Blakes et al., 2021*).

Several observations indicate that the described role of the αI-helix in the destabilization of the regulatory core is incomplete. First, the αI-helix is rather flexible in solution (*Jahnke et al., 2010*; *Skora et al., 2013*) and may avoid the steric clash with the SH2 domain in the assembled conformation. Indeed, an Abl[83-534] construct comprising only the SH3-SH2-KD domains but not the myristoylated N-cap adopts an assembled conformation in solution (*Skora et al., 2013*) and has strongly reduced enzymatic activity relative to the isolated KD (*Sonti et al., 2018*). Second, high-affinity binding of type II ATP site inhibitors to the ATP site (*Grzesiek et al., 2022*; *Sonti et al., 2018*), which induce an inactive activation loop (A-loop) conformation in the KD, disassembles the core into an arrangement where SH3 and SH2 domains move with high-amplitude nanosecond motions relative to the KD (*Skora et al., 2013*; *Figure 1D*). This type II inhibitor-disassembled core is reassembled when allosteric inhibitors bind to the myristate pocket (*Skora et al., 2013*), indicating a cross talk between ATP site and the αI-helix/myristate pocket. The inhibitor-induced disassembly correlates with a slight rotation of the KD N-lobe toward the SH3 domain caused by the push of type II inhibitors onto the

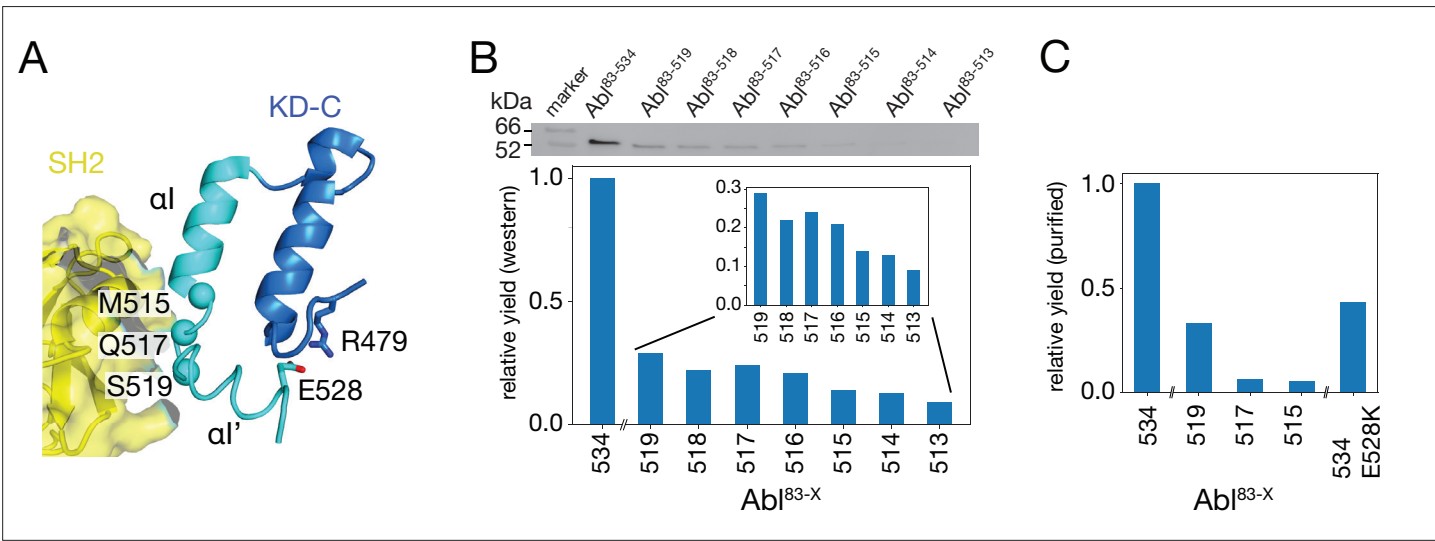

**Figure 2.** Production of C-terminally truncated Abl regulatory core constructs. (**A**) Detailed view of the interaction site between the SH2 domain (yellow) and the C-terminus of the Abl KD with its αH- (dark blue) and αI- (cyan) helices from the crystal structure of the assembled Abl core (PDB 2FO0). The Cα-atoms of M515, Q517, and S519 are shown as cyan spheres (**B**) Top: western blot of *E. coli*-expressed, soluble truncated Abl constructs in the supernatant after cell lysis. Bottom: expression yields of the corresponding Abl constructs as quantified from the gel bands. The numbers on the horizontal axis denote the last amino acid of the respective construct Abl[83-X]. (**C**) Relative yields of the purified truncated Abl constructs and Abl[83-534,E528K] as quantified by OD[280].

The online version of this article includes the following source data for figure 2:

**Source data 1.** Uncropped original image file of western blot in *Figure 2B*.

**Source data 2.** Uncropped western blot of *Figure 2B* with annotation of relevant bands.

A-loop and subsequently the P-loop (*Sonti et al., 2018*). To explain these observations, we have proposed a mechanical model where both the rotation of the KD N-lobe and the flexible αI-helix exert destabilizing forces onto the SH3/SH2-KD interface, which together lead to the disassembly of the core (*Sonti et al., 2018*).

Here, we prove and refine this model of allosteric activation and interaction between the αI-helix and the ATP site by stepwise shortening the Abl KD C-terminus from residue 534 to residue 513. We show that αI-helix truncations beyond residue 519, namely removal of the αI–αI′ loop reduce both the imatinib-induced disassembly of the Abl regulatory core and its kinase activity, which confirms the cross talk between αI/SH2 and KD N-lobe/SH3 interfaces. A salt bridge between E528 in the αI′-helix and R479 on the KD C-lobe appears as a crucial element of Abl core regulation. Its disruption by the malignant mutation E528K results in a strongly increased kinase activity due to the unlocking of the αI′-helix. A simple mechanical model of Abl regulation involving the two forces from the KD N-lobe and the αI′-helix onto the KD/SH3SH2 interface explains all current observations.

## Results and discussion

### Expression of C-terminal Abl truncation constructs

To test the impact of the C-terminal αI-helix on Abl's conformational equilibria and activity, Abl constructs of the regulatory core (Abl[83-534]) were expressed with decreasing αI-helix length by introducing stop codons (*Figure 2*). While Abl constructs with partial truncations of the αI′-helix (Abl[83-522] and Abl[83-525]) could not be purified to homogeneity and were very unstable, the first well purifiable truncation construct was Abl[83-519], which misses the entire αI′-helix part of the myristoyl-bound conformation (*Figure 2A*). As compared to Abl[83-534], its expression yield is reduced about fourfold as quantified by western blot (*Figure 2B*) and analysis of the fully purified construct (*Figure 2C*). The C-terminus was then further truncated by one amino acid at a time until Abl[83-513], thereby removing sequentially the αI–αI′ loop and the last two residues T514 and M515 of the αI-helix part of the myristoyl-bound conformation. The resulting constructs had further reduced expression yields down to less than 10% for the least expressing Abl[83-513] relative to Abl[83-534] (*Figure 2B*). This reduction in yield can be rationalized by the sequential loss of stabilizing interactions between the αI-αI′ loop and the αI′-helix involving residues F516, Q517, S520, and D523 (*Nagar et al., 2003*; *Nagar et al., 2006*) and the loss of α-helical backbone hydrogen bonds originating at T514 and M515.

Due to these severe reductions in yield, large-scale expressions of [15]N-labeled constructs for NMR and kinase assays were then only carried out for Abl[83-519], Abl[83-517], and Abl[83-515]. The final yield after purification for Abl[83-515] was 0.16 mg per liter expression culture and reduced about 20-fold relative to Abl[83-534] (3.0 mg/L; *Figure 2C*). Furthermore, also a full-length Abl[83-534] construct harboring the malignant E528K mutation could be purified with good yield (1.3 mg/L).

### Truncation of the C-terminal αI–αI′-loop leads to less disassembly of the Abl core by imatinib

Assembled and disassembled Abl core conformations can be readily distinguished by the [1]H-[15]N chemical shifts of the SH3 and SH2 domain backbone resonances (*Skora et al., 2013*; *Sonti et al., 2018*). The well-resolved resonances of V130, T136, and G149 shift particularly strongly (*Figure 3A*, left) when the type II inhibitor imatinib binds to Abl[83-534], which leads to the disassembly of the Abl core as shown by SAXS, NMR relaxation and RDC data (*Skora et al., 2013*). No pronounced chemical shift changes are observed for these resonances between the Abl[83-534]•imatinib and the Abl[83-519]•imatinib complexes, indicating that the latter is also disassembled. Upon further truncation of the Abl C-terminus (Abl[83-517] and Abl[83-515]), the resonances of the imatinib complexes shift on a straight line from the disassembled Abl[83-534]•imatinib in the direction of the Abl[83-534] apo form, which is in the assembled conformation (*Skora et al., 2013*). In contrast, the apo forms of the Abl C-terminal truncations have very similar, yet not completely identical, chemical shifts to apo Abl[83-534] (*Figure 3A*, center) which indicates that they are also assembled (see below).

A full quantitative analysis of the observed SH3 and SH2 [1]H-[15]N chemical shifts for all Abl constructs in apo form and in complex with imatinib by Principal Component Analysis (PCA) is shown *Figure 3B and C*. The PCA also includes previously observed chemical shifts for binary Abl[83-534] complexes with 5 type II, 8 type I ATP site inhibitors, AMP-PNP, as well as with the allosteric inhibitors GNF-5 and

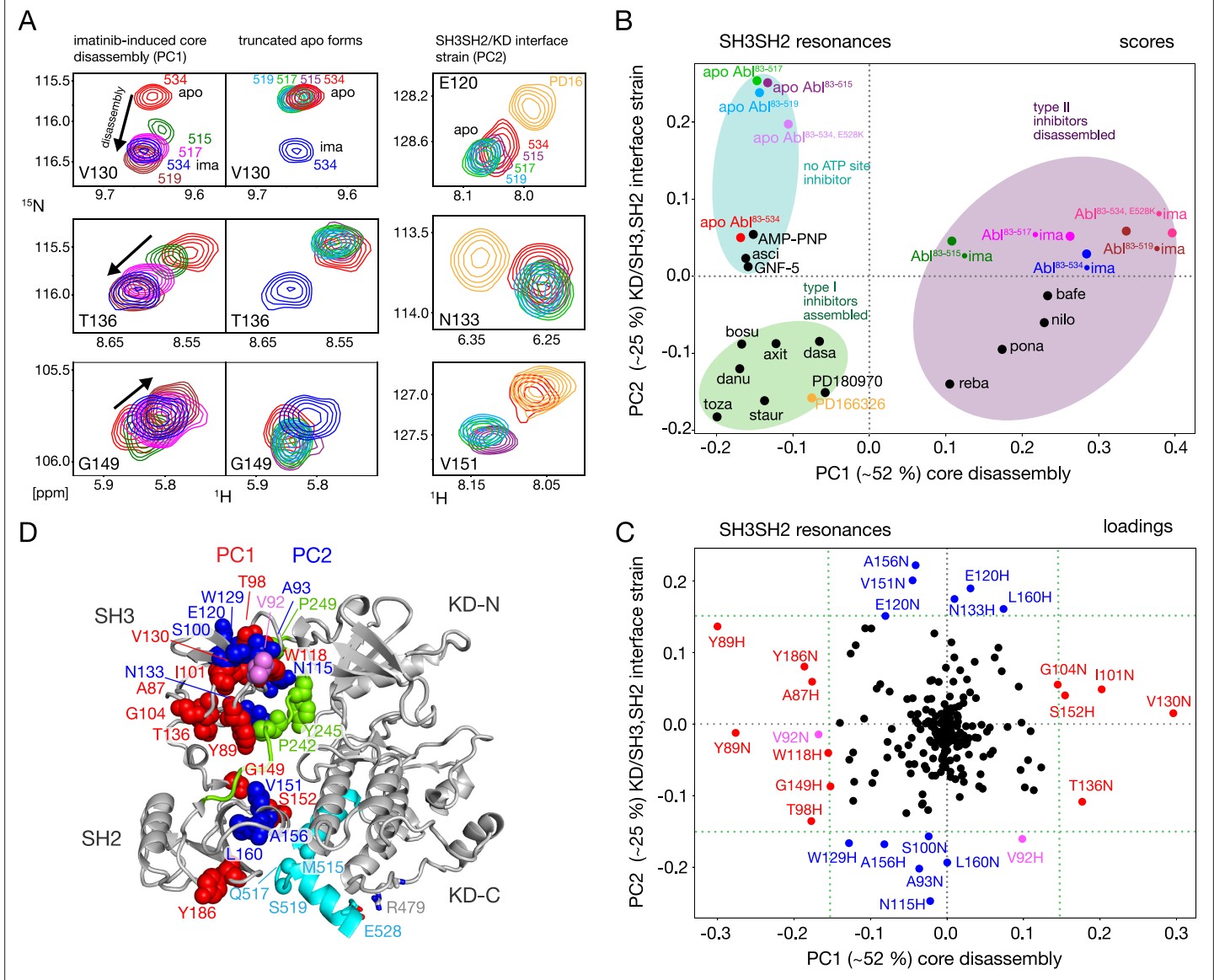

**Figure 3.** NMR evidence for reduced imatinib-induced core disassembly in truncated Abl constructs. (**A**) Selected ¹H-¹⁵N TROSY resonances that report on the imatinib-induced disassembly (left and middle panels) and the SH3SH2/KD interface (right panel). The label indicates the last amino acid of the respective Abl[83-X] construct. Abbreviations: ima:imatinib, PD16:PD166326.(**B**) Scores plot of the PCA of the SH3 and SH2 domain chemical shifts of various Abl core complexes and apo forms. Abbreviations: asci:asciminib, bosu:bosutinib, axi:axitinib, dasa:dasatinib, danu:danusertib, toza:tozasertib, staur:staurosporine, reba:rebastinib, pona:ponatinib, nilo:nilotinib, bafe:bafetinib. (**C**) Loadings plot of the PCA shown in panel B indicating contributions of individual ¹Hᴺ and ¹⁵N chemical shifts to the first two PCs. Resonances with absolute PC1 or PC2 loadings larger than 0.15 are highlighted in red and blue, respectively. V92 has large loadings for both PCs and is shown in pink. (**D**) Residues with large PC1 or PC2 loadings according to panel C indicated as spheres within the assembled Abl core structure (PDB 2FO0). The color coding follows panel C.

asciminib (***Sonti et al., 2018***). The PCA comprises 113 SH3-SH2 distinct ¹H-¹⁵N resonances (out of a total of 148 non-proline residues). The combined first (PC1) and second (PC2) principal components explain 77% of these data.

As observed previously (***Sonti et al., 2018***), the PC1 scores (***Figure 3B***) readily distinguish between assembled and disassembled conformations. All type II inhibitor complexes cluster in one PC1 score region corresponding to disassembled conformations, whereas complexes with type I inhibitors, allosteric inhibitors, and AMP-PNP as well as the full-length apo form cluster in a second PC1 score region corresponding to assembled conformations. Notably, all truncated apo forms also fall in the latter PC1 score region, corroborating clearly that these are also in the assembled conformation. The

assembled conformations are further differentiated by the PC2 scores into a region comprising all type I inhibitor complexes and a second region containing all apo forms that have no inhibitor in the ATP site, that is the allosteric inhibitor and AMP-PNP complexes.

The PCA loadings (*Figure 3C*) show the contributions of individual residue chemical shift changes to PC1 and PC2. Residues with |PC1|≥0.15, which are strongly affected by the core assembly-disassembly transition, are annotated in red and indicated as red spheres in the assembled Abl core structure (*Figure 3D*). While they are distributed to some extent across both SH3 and SH2 domains, residues G104, T136, Y89, and G149 form a notable cluster at the SH3/SH2 interface. This is consistent with the altered environment of residues in this region expected from the disassembly of the regulatory core and the increased interdomain flexibility (*Skora et al., 2013*). Residues strongly contributing to PC2 (|PC2|≥0.15) are marked in blue in *Figure 3C and D*. Previously, the largest PC2 score differences of Abl[83-534] were observed between the Abl[83-534]•type I inhibitor complexes and apo Abl[83-534] (*Sonti et al., 2018*). Interestingly, now the truncated apo Abl constructs shift even further away from the type I inhibitor complexes than apo Abl[83-534], but show little difference among their different truncation lengths. In contrast to PC1, residues contributing most to PC2 are located either in the upper part of the SH3 domain toward the KD N-lobe or the upper part of the SH2 domain toward the KD αI–αI' loop (*Figure 3D*). Both regions are also in direct contact with the KD-SH2 linker (green, *Figure 3D*). As the affected residues react differently to perturbations by type I inhibitors and truncation of the αI'-helix (*Figure 3A*, right), we attribute this behavior to two effects intermixed into the PC2 detection: (i) a minor rearrangement of the SH3/KD N-lobe interface caused by filling of the ATP pocket with type I inhibitors, which in contrast to the stronger N-lobe motion induced by type II inhibitors does not yet lead to core disassembly and (ii) a small rearrangement of the SH2/KD C-lobe interface caused by shortening and mutations of the αI-helix.

## Conformational changes within KD and SH2 domain induced by the αI-helix truncation

The PCA analysis shown in *Figure 3* was restricted to the SH3 and SH2 domains and did not include the KD, since the $^1$H-$^{15}$N resonances of the former are best suited to monitor the Abl regulatory core assembly-disassembly transition, while not being affected by local changes from ligand binding. An analogous PCA analysis of the $^1$H-$^{15}$N KD resonances (164 out of a total of 270 non-proline amino acids) for all investigated Abl constructs and complexes reveals details of these local ligand-induced structural changes as well as of the effects of the αI-helix truncations. Similar to the analysis of the SH2 and SH3 resonances, the PC1/PC2 scores plot of the KD resonances (*Figure 4A*) clearly separates type I and type II ATP site inhibitor complexes, as well as all forms without any ATP site inhibitor. The respective PC1/PC2 loadings (*Figure 4—figure supplement 1A*) show that the amino acids, which cause this separation by their pronounced shifts, are distributed across the entire KD. However, effects from the ATP site, allosteric site binding, and the αI-helix truncation are strongly intermixed in this analysis, and specific causes are hard to distinguish.

To separate these effects, two additional PCAs were carried out. The first was restricted to the KD resonances of only type I and type II inhibitor-bound forms (*Figure 4B and C*). The PC1 scores of this analysis separate the two types of inhibitors in a very clear manner (*Figure 4B*). The respective PC1 loadings (*Figure 4C*, *Figure 4—figure supplement 1*) show that the affected residues are mostly adjacent to the A-loop in its inactive conformation, which is observed in the type II inhibitor-bound crystal structures, as well as adjacent to the nearby P-loop. Thus, these residues evidently feel a force when type II inhibitors bind. This finding corroborates the previously proposed mechanism for type II inhibitor-induced core disassembly (*Sonti et al., 2018*), namely that type II inhibitors push onto the A-loop and subsequently the P-loop thereby rotating the KD N-lobe toward the SH3 domain and exerting a destabilizing force onto the SH3/KD N-lobe interface.

A further PCA was calculated on all observed SH3-SH2-KD resonances of all apo forms (*Figure 4D and E*). This analysis of the entire apo SH3-SH2-KD protein is meaningful, since the KD resonances are not dominated by the strong effects of ATP site binders. The PC1 scores (*Figure 4D*) separate wild-type apo Abl[83-534] from all the truncated Abl apo forms (Abl[83-515], Abl[83-517], Abl[83-519]) and the Abl[83-534,E528K] mutant. The respective PC1 loadings of significant size (*Figure 4E*, *Figure 4—figure supplement 1* |PC1|>0.1) are mostly from amino acids in the vicinity of the αI-helix and myristoyl pocket, indicating localized structural changes in this region due to the mutations of the αI-helix.

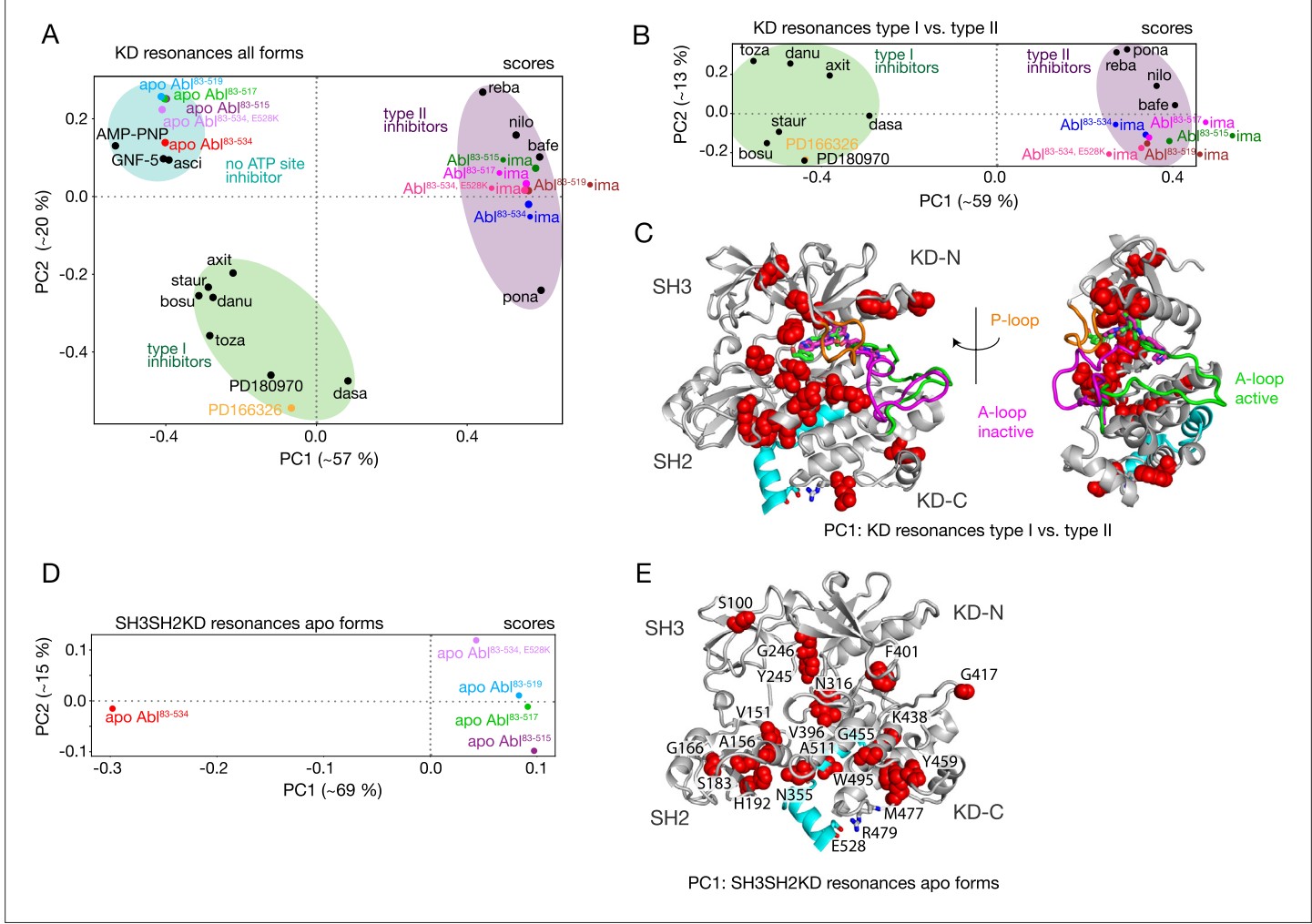

**Figure 4.** PCAs of resonances comprising the KD or entire Abl core. (**A**) PCA scores of the KD resonances of all complexes shown in *Figure 3B*. (**B**) PCA scores of the KD resonances of only type I and type II inhibitor complexes. The respective loadings are shown in *Figure 4—figure supplement 1*. (**C**) Residues with absolute PC1 loadings larger than 0.15 for the analysis in panel B are shown as red spheres in the assembled core structure (PDB 2FO0). The P-loop is shown in orange and the A-loop in its active (green, PDB 2FO0) as well as inactive (magenta, PDB 2HYY *Cowan-Jacob et al., 2007*) conformation. (**D**) PCA scores of the resonances of the entire Abl core for the apo forms of all αI-helix variants. The respective loadings are shown in *Figure 4—figure supplement 1*. (**E**) Residues with absolute PC1 loadings larger than 0.1 of the analysis in panel D are shown as red spheres in the assembled core structure (PDB 2FO0).

The online version of this article includes the following figure supplement(s) for figure 4:

**Figure supplement 1.** PCA loadings showing the contributions of the $^1$H and $^{15}$N chemical shifts of individual residues to PC1 and PC2.

## The αI-helix contributes directly to Abl activation

The NMR data show that the αI-helix plays a crucial role in the imatinib-induced Abl core disassembly and exerts forces onto the SH2-SH3 to KD interface. To quantify the role of the αI-helix in the regulation of Abl, we assayed the kinase activity of the different truncation constructs in vitro using the optimized Abltide peptide substrate (*Figure 5A and B*, *Table 1*). Indeed, the stepwise truncation of the αI-helix continuously decreased the Michaelis-Menten $v_{max}$ (*Figure 5A and B*) for the different helix lengths reaching a minimum of 3.2 nmol $P_i$ min$^{-1}$µmol$^{-1}$ for Abl$^{83-515}$, which is about 30 times smaller than the wild-type Abl$^{83-534}$ value (89.4 nmol $P_i$ min$^{-1}$µmol$^{-1}$). In contrast, the substrate dissociation constant $K_M$ remained almost constant within the error limits (~40–90 µM). The reduction of the enzymatic activity by the αI truncations is consistent with the notion that a shorter αI-helix favors the assembled conformation of the Abl regulatory core, thereby hindering 'breathing' motions between the KD N- and C-lobes and impeding substrate binding. Conversely, the full-length αI-helix exerts

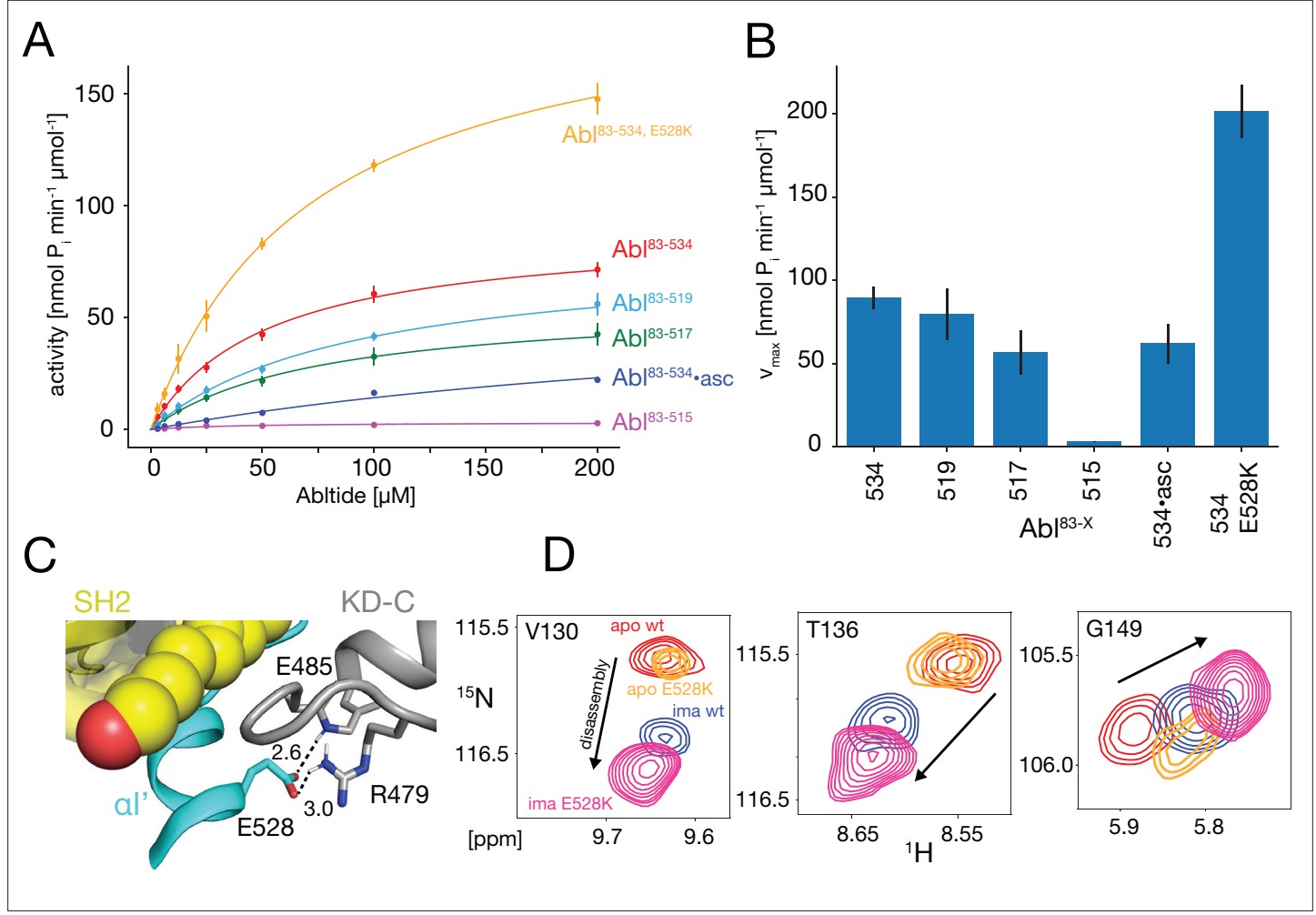

**Figure 5.** In vitro kinase activity assays and effect of the E528K mutation. (**A**) Specific kinase activity of Abl$^{83-534, E528K}$ (orange), Abl$^{83-534}$ (red), Abl$^{83-519}$ (light blue), Abl$^{83-517}$ (green), Abl$^{83-515}$ (magenta), and Abl$^{83-534}$•asciminib (asc, blue). Sample sizes and statistical information are given in **Table 1**. (**B**) Bar plot of fitted Michaelis-Menten $v_{max}$ values derived from the kinase assays in panel A. Numerical values are given in **Table 1**. (**C**) Detailed view of the E528-R479 salt bridge in the structure of the assembled Abl core (PDB 2FO0). Distances between the E528 carboxylate oxygens and the guanidinium sidechain hydrogen of R479 as well as the backbone amide hydrogen of E485 are displayed as dashed lines and indicated in Angstrom. (**D**) Superpositions of the $^{1}$H-$^{15}$N TROSY resonances of V130, T136, and G149 for apo Abl$^{83-534,wt}$ (red), apo Abl$^{83-534,E528K}$ (orange), Abl$^{83-534,wt}$•imatinib (blue), and Abl$^{83-534,E528K}$•imatinib (magenta) showing the increased imatinib-induced core disassembly of the Abl$^{83-534,E528K}$ mutant.

**Table 1.** Michaelis-Menten parameters[*] for the various Abl helix constructs.

|  | Abl$^{83-534}$ | Abl$^{83-519}$ | Abl$^{83-517}$ | Abl$^{83-515}$ | Abl$^{83-534}$•asciminib | Abl$^{83-534,E528K}$ |
|---|---|---|---|---|---|---|
| $v_{max}$[†] | 89.4±6.7 | 79.5±15.4 | 56.8±13.3 | 3.2±0.2 | 61.9±11.8 | 201.7±16.0 |
| $K_M$[‡] | 51.8±8.8 | 91.4±30.9 | 74.8±35.7 | 39.5±9.0 | 336.7±80.3 | 71.6±12.3 |
| N[§] | 8 | 3 | 3 | 3 | 3 | 2 |

[*]the Michaelis-Menten parameters and their errors were obtained by Monte Carlo fitting using the sample mean and standard error of the mean of the measured kinase activities shown in **Figure 5A**.

[†]in nmol P$_i$ min$^{-1}$µmol$^{-1}$, where P$_i$ is the transferred phosphate.

[‡]in µM.

[§]total number of activity assay experiments.

forces onto the SH3-SH2 interface that either increase KD N-/KD C-lobes hinge motions or lead to complete core disassembly, thereby increasing the enzymatic activity.

## Glutamate 528 is a crucial stabilizer for the autoinhibited conformation

The E528K mutation, located toward the end of helix αI', has been found in patients with the recently described ABL1 malformation syndrome (*Blakes et al., 2021*). An inspection of the assembled Abl core structure reveals a potential salt bridge between E528 and R479 in the KD C-lobe adjacent to the myristoyl pocket (*Figure 5C*). The malignant E528K mutation is expected to disrupt this salt bridge. We speculated that this salt bridge may be a stabilizing element, which restricts the αI' motion and prevents forces toward the SH2 domain that could lead to core disassembly and higher activity. Albeit we could not study the effect of the αI'-helix length beyond residue 519 due to the instability of the respective truncation mutants, it was possible to express the Abl[83-534,E528K] mutant. Indeed, the chemical shifts of the Abl[83-534,E528K] SH2 and SH3 domains show that this mutant is more disassembled

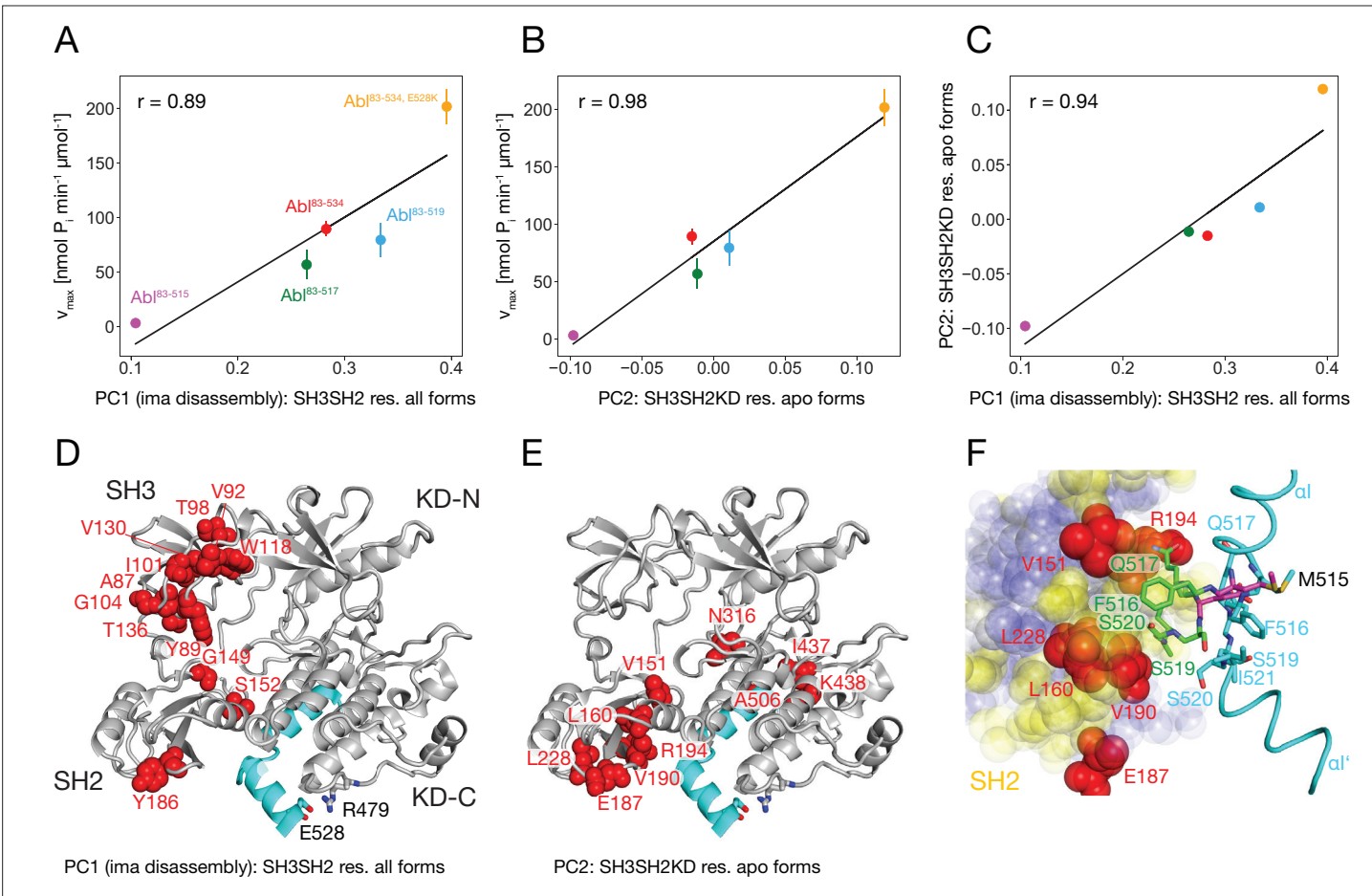

**Figure 6.** Correlation between Abl kinase activity, imatinib-induced Abl core disassembly, and αI-helix mutations. (**A**) Correlation between the PC1 scores of the SH3SH2 resonances (PCA in *Figure 3B*) of all imatinib complexes and the respective kinase activity (*Figure 5B*). (**B**) Correlation between the PC2 scores of the SH3SH2KD resonances of all apo forms (PCA in *Figure 4D*) and the respective kinase activity (*Figure 5B*). The color code for the Abl constructs follows panel A. (**C**) Correlation between the PC1 scores of the SH3SH2 resonances of all imatinib complexes (*Figure 3B*) and the PC2 scores of the SH3SH2KD resonances of all apo forms (*Figure 4D*). The color code for the Abl constructs follows panel A. (**D**) Residues with absolute PC1 loadings larger than 0.15 from the PCA of the SH3SH2 resonances of all imatinib complexes (*Figure 3C*) indicated as red spheres within the structure of the assembled Abl core (PDB 2FO0). (**E**) Residues with absolute PC2 loadings larger than 0.1 from the PCA of the SH3SH2KD resonances of all apo forms (*Figure 4D*, *Figure 4—figure supplement 1*) indicated as red spheres within the structure of the assembled Abl core. (**F**) Interface between the αI-helix and the SH2 domain. The bent αI-helix of the assembled core structure is shown in cyan. The straight αI-helix of the isolated Abl kinase domain (PDB 1M52) is shown in magenta stick representation, while residues that clash with the SH2 domain of the assembled Abl core being shown in green stick representation. SH2 residues are displayed as colored spheres. Red: residues with absolute PC2 loadings >0.1 as in panel E, blue: residues with absolute PC2 loadings smaller than 0.1, yellow: residues with unresolved resonances in NMR spectrum.

upon imatinib binding than Abl[83-534] (*Figures 3B and 5D*). In addition, the kinase activity ($v_{max}$) of the Abl[83-534,E528K] mutant was increased more than two-fold relative to Abl[83-534], while $K_M$ remained unaffected (*Figure 5A and B*, *Table 1*). Thus, as expected, the disruption of the E528-R479 salt bridge significantly increases both Abl's enzymatic activity as well as the tendency for Abl core disassembly.

## Correlation between imatinib-induced Abl core disassembly, apo conformation, and apo kinase activity

It is striking that for all investigated αI-helix mutants, the kinase activity and the tendency of imatinib to open the assembled core correlate. This can be quantified by comparing the $v_{max}$ of these various Abl mutants and the respective principal component 1 (PC1) score of the SH2/SH3 domain chemical shifts in their imatinib-bound forms, which is a measure of the core disassembly (*Figure 3B*). Indeed, the correlation between $v_{max}$ and the PC1 score is highly significant with a Pearson r of 0.89 (*Figure 6A*). Thus, the imatinib-induced core disassembly and the activating motions required for kinase activity seem to be controlled by a common mechanism, which originates at the αI-helix. A thorough inspection of the PCA results for all SH3SH2KD resonances of the apo αI-helix mutants (*Figure 4D*) revealed that its PC2 score also follows the kinase activity. A quantitative correlation revealed a highly significant Pearson r of 0.98 between this PC2 score and $v_{max}$ (*Figure 6B*). Thus, the chemical shift changes captured by the PC2 of the apo αI mutants report structural variations which are related to the changes in the kinase activity. The strongest respective chemical shift changes (according to the PC2 loadings, *Figure 6E*) occur at the interface of the SH2 domain (V151, L160, E187, V190, R194, L228) with the KD C-lobe and in the vicinity of the αI N-terminal end (N316, I437, K438). Clearly, as both NMR resonance PC scores (PC1 imatinib-bound forms and PC2 apo forms) correlate to the kinase $v_{max}$, they must also correlate to each other, and a respective high Pearson r of 0.94 is observed (*Figure 6C*). Taken together, these results show that the truncations and mutations of the αI-helix exert forces from the αI-helix onto the SH2 domain, which modulate both the kinase activity and the imatinib-induced opening of the core.

To obtain more insights into these forces we carefully compared the assembled core (myristoylated and in complex with the type I inhibitor PD166326) structure with the bent αI-helix and the isolated Abl kinase domain (in complex with the type I inhibitor PD173955) structure with the straight αI-helix (*Figure 6F*). A superposition reveals that steric clashes of the straight αI-helix with the SH2 domain are mostly expected for residues F516, Q517, S520, and I521 residing in the αI–αI' loop. In the assembled core structure, these residues are within van-der-Waals distance of several SH2 residues, for which no well-resolved resonances could be detected in the NMR spectra (yellow residues in *Figure 6F*). However, the latter are in direct contact with the SH2 residues previously identified as having the strongest chemical shift PC2 loadings caused by the mutations of the αI-helix (red residues in *Figure 6E and F*). These results establish that the most significant part of the forces from the αI-helix onto the SH2 domain is transmitted from the αI–αI' loop toward a region on the SH2 surface, which is centered around residue V190.

## Mechanics of Abl kinase regulation by the αI'-helix

Based on the current data we can refine our previous mechanical model of Abl regulation (*Sonti et al., 2018*), which is based on the assumption that a disassembled regulatory core is necessary for high activity (*Figure 7*). Two main mechanical forces drive the opening of the core: $F_{KD-N,SH3}$ acts between the KD N-lobe and the SH3 domain and $F_{αI,SH2}$ acts between the αI-helix and the SH2 domain.

In the absence of ATP site ligands, the KD N-lobe wobbles moderately within the closed regulatory core as evident from conformational exchange observed by NMR (*Skora et al., 2013*). This wobbling exerts a moderate force $F_{KD-N,SH3}$ onto the KD N-lobe/SH3 interface. A high flexibility of the αI'-helix has been observed in the absence of myristoyl pocket binders in a KD construct (*Jahnke et al., 2010*). Dynamic motions or static steric clashes of the αI-helix generate a force $F_{αI,SH2}$ onto the SH2 domain. The combined effect of $F_{KD-N,SH3}$ and $F_{αI,SH2}$ in the absence of any ligands lead to an occasional opening of the regulatory core and consequently moderate kinase activity (*Figure 5A*).

When type II, but not type I inhibitors bind to the ATP pocket, they exert pressure onto the A-loop and P-loop which rotates the entire N-lobe by 5–10° toward the SH3 domain. This rotation is observed in various crystal structures (*Sonti et al., 2018*). The rotation increases the force $F_{KD-N,SH3}$ and leads to core disassembly. However, when allosteric inhibitors bind to the myristoyl pocket, they reduce the αI'-helix dynamics (*Jahnke et al., 2010*) and thereby the force $F_{αI,SH2}$. This then leads to core

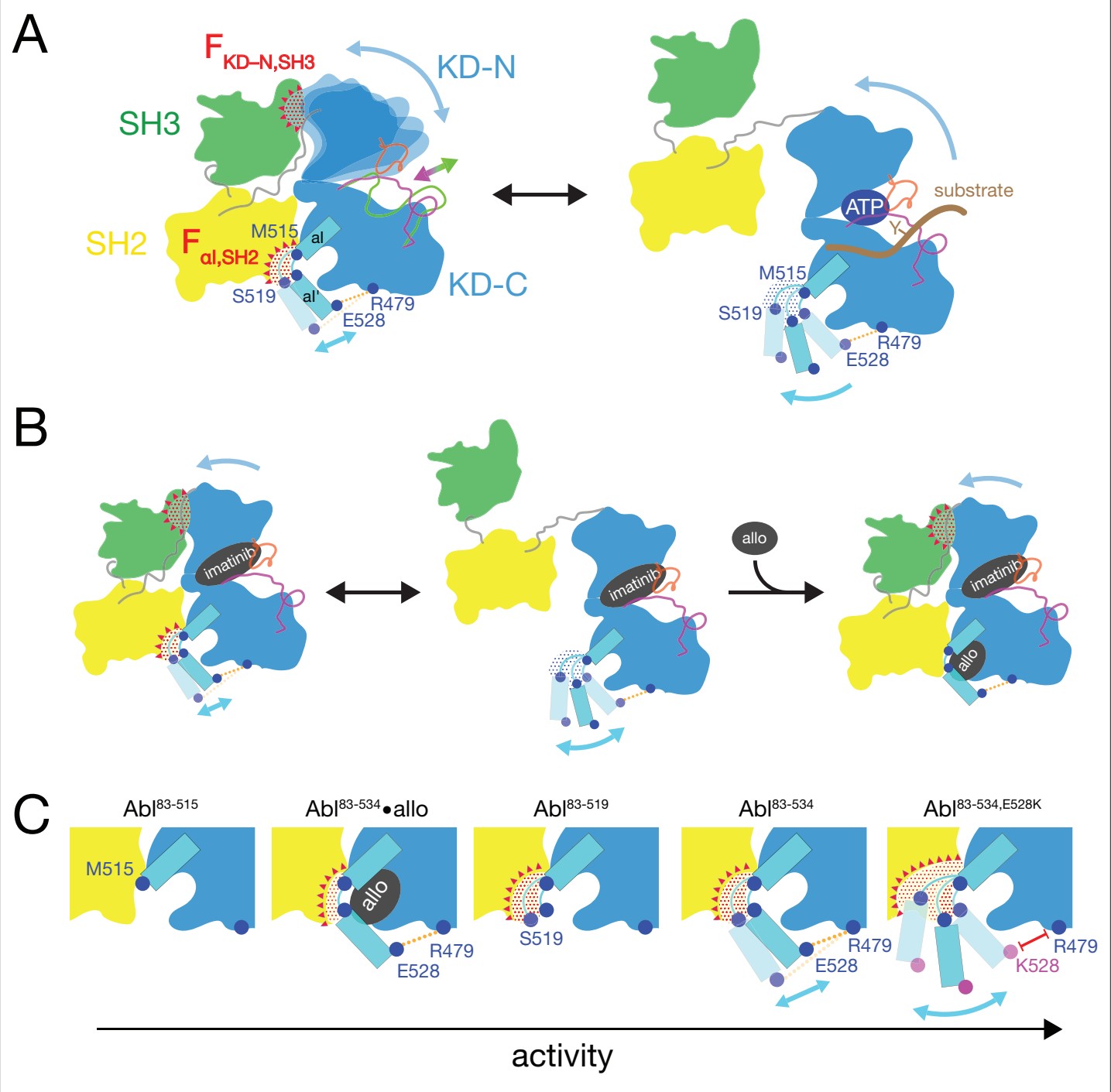

**Figure 7.** Mechanical model of Abl allosteric regulation. (**A**) Transition between closed (left) and open (right) conformation of Abl's regulatory core during activation. Arrows indicate mobility of the KD N-lobe and the αI-helix. Clashes between SH3 and KD-N and between SH2 and αI-helix resulting from this mobility are shown by red dots, the related clash-induced forces $F_{KD-N,SH3}$ and $F_{\alpha I,SH2}$ by red triangles. Color coding: SH3 (green), SH2 (yellow), KD (blue), αI-helix (cyan), A-loop (inactive: magenta, active: green), P-loop (orange), tyrosine substrate (brown), R479-E528 salt bridge (orange). (**B**) Model of imatinib-induced Abl core disassembly. The color code follows panel A. Allo: allosteric inhibitor. (**C**) Detailed model of the interaction between the αI-helix and the SH2 domain for the various investigated Abl αI-helix constructs and inhibitor-bound conformations. The observed, respective Abl activity increases in the direction of the arrow at the bottom. The color code follows panel A.

reassembly in the presence of type II inhibitors. In the absence of ATP site inhibitors, the reduction of the force $F_{\alpha I,SH2}$ by the allosteric inhibitors additionally stabilizes the assembled core and further reduces kinase activity (*Figure 5A*).

The data of the various truncations and mutations of the αI'-helix provide the following detailed picture of its role in Abl activation (*Figure 7C*). In wild-type apo Abl[83-534], the αI'-helix motion is apparently restricted by the E528-R479 salt bridge. This is evident from the fact that the disruption of the salt bridge in the Abl[83-534,E528K] mutant leads to strongly increased activity (*Figure 5A*) and a stronger core disassembly by imatinib (*Figures 3B, 5D and 6A*). Both effects can be explained by an increased force $F_{\alpha I,SH2}$ acting onto the SH2 domain, which is evident by the increased PC2 score of apo Abl[83-534,E528K] (*Figures 4D and 6B*). Nevertheless, also the αI-helix of wild-type apo Abl[83-534] exerts a force $F_{\alpha I,SH2}$ onto the SH2 domain. The latter is reduced in the presence of certain myristoyl pocket binders such as asciminib, which change the αI-αI' loop conformation and lock the αI'-helix onto the KD C-lobe. The $F_{\alpha I,SH2}$ force of the wild-type αI-helix apparently does not mainly originate from its αI' part, since the truncation Abl[83-519] has no strong effect. It neither significantly reduces the imatinib-induced core disassembly, nor the apo activity, nor the apo PC2 score, that is the force $F_{\alpha I,SH2}$ (*Figure 6A and B*). However, the further truncation Abl[83-515], which removes the αI-αI' loop, abolishes the kinase activity almost completely (*Figure 5A and B*) and strongly decreases the $F_{\alpha I,SH2}$ force as well as the imatinib-induced core disassembly (*Figure 6A and B*). This indicates that the αI helix of wild-type Abl mainly exerts the $F_{\alpha I,SH2}$ force onto the SH2 domain via the αI-αI' loop. The αI-αI' loop is flexible to a certain extent since various conformations are observed in the presence and absence of myristoyl pocket binders. Presumably, the $F_{\alpha I,SH2}$ force is exerted by steric clashes of some members from the conformational ensemble of the flexible loop.

## Conclusion

The allosteric connection between Abl ATP site and myristate site inhibitor binding has been noted before, albeit specific settings such as construct boundaries and the control of phosphorylation vary in published experiments. Positive and negative binding cooperativity of certain ATP-pocket and allosteric inhibitors has been observed in cellular assays and in vitro (*Kim et al., 2023*; *Zhang et al., 2010*). Furthermore, hydrogen exchange mass spectrometry has indicated changes around the unliganded ATP pocket upon binding of the allosteric inhibitor GNF-5 (*Zhang et al., 2010*). Here, we present a detailed high-resolution explanation of these allosteric effects via a mechanical connection between the kinase domain N- and C-lobes that is mediated by the regulatory SH2 and SH3 domains and involves the αI helix as a crucial element.

Specifically, we have established a firm correlation between the kinase activity of the Abl regulatory core, the imatinib (type II inhibitor)-induced disassembly of the core, which is caused by a force $F_{KD-N,SH3}$ between the KD N-lobe and the SH3 domain, and a force $F_{\alpha I,SH2}$ exerted by the αI-helix toward the SH2 domain. The $F_{\alpha I,SH2}$ force is mainly caused by a clash of the αI-αI' loop with the SH2 domain. Both the $F_{KD-N,SH3}$ and $F_{\alpha I,SH2}$ force act on the KD/SH2SH3 interface and may lead to the disassembly of the core, which is in a delicate equilibrium between assembled and disassembled forms. As disassembly is required for kinase activity, the modulation of both forces constitutes a very sensitive regulation mechanism. Allosteric inhibitors such as asciminib and also myristoyl, the natural allosteric pocket binder, pull the αI-αI' loop away from the SH2 interface, and thereby reduce the $F_{\alpha I,SH2}$ force and activity. Notably, all observations described here were obtained under non-phosphorylated conditions, as phosphorylation will lead to additional strong activating effects.

The presented mechanical force model also explains the malignance of the αI'-helix E528K mutation in the recently described ABL1 malformation syndrome. The native E528 forms a salt bridge with R479 in the KD C-lobe, which stabilizes the inactive, assembled conformation by reducing the conformational space of the αI'-helix, and thereby the $F_{\alpha I,SH2}$ force. In contrast, the disruption of the E528-R479 salt bridge by the E528K mutation strongly increases the kinase activity.

We have used here imatinib binding to the ATP-pocket as an experimental tool to disassemble the Abl regulatory core. Our previous analysis (*Sonti et al., 2018*) of the high-resolution Abl transition-state structure (*Levinson et al., 2006*) indicated that due to the extremely tight packing of the catalytic pocket, binding and release of the ATP and tyrosine peptide substrates is only possible if the P-loop and thereby the N-lobe move toward the SH3 domain by about 1–2 Å. This motion is of similar size and direction as the motion of the N-lobe observed in complexes with imatinib and other type II inhibitors (*Sonti et al., 2018*). From this we concluded that substrate binding opens the Abl core in a

similar way as imatinib. The present NMR and activity data now clearly establish the essential role of the αI-helix both in the imatinib- and substrate-induced opening of the core, thereby further corroborating the similarity of both disassembly processes.

Notably, the used regulatory core construct Abl[83-534] lacks the myristoylated N-cap. Although we have previously demonstrated that the latter construct is predominantly assembled (*Skora et al., 2013*), the addition of the myristoyl moiety is expected to further stabilize the assembled conformation in a similar way as asciminib. Considering this mechanism, dissociation of myristoyl from the native Abl 1b core may be a first step during activation. However, it should be kept in mind that the Abl 1 a isoform lacks the N-terminal myristoylation, and it is presently unclear whether other moieties bind to the myristoyl pocket of Abl 1 a during cellular processes.

# Materials and methods

**Key resources table**

| Reagent type (species) or resource | Designation | Source or reference | Identifiers | Additional information |
|---|---|---|---|---|
| Gene (Human) | Abl | Uniprot | P00519-2 | |
| Gene (*Escherichia phage lambda*) | Lambda phosphatase; LPP | Uniprot | P03772 | |
| Strain, strain background (*Escherichia coli*) | BL21(DE3) | Sigma-Aldrich | CMC0014 | Chemical competent cells |
| Recombinant DNA reagent | Plasmid containing human Abl[83-534] | *Skora et al., 2013* DOI:10.1073/pnas.1314712110 | | |
| Recombinant DNA reagent | Plasmid containing Lambda phosphatase | *Sonti et al., 2018* DOI:10.1021/jacs.7b12430 | | |
| Antibody | Conjugated poly-histidine antibody | Sigma-Aldrich | Cat.# A7058 | Mouse monoclonal (1:10000); peroxidase conjugate |
| Peptide, recombinant protein | Abltide | ProteoGenix | | biotin-GGEAIYAAPFKK |
| Commercial assay or kit | SAM2 Biotin Capture Membrane | Promega | Cat.# V2861 | |
| Chemical compound, drug | Asciminib; asci | Selleck Chemicals | Cat.# S8555 | |
| Chemical compound, drug | GNF-5 | Selleck Chemicals | Cat.# S7526 | |
| Chemical compound, drug | AMP-PNP | Roche | Cat.# 10102547001 | |
| Chemical compound, drug | Bosutinib; bosu | Selleck Chemicals | Cat.# S1014 | |
| Chemical compound, drug | Axitinib; axit | Selleck Chemicals | Cat.# S1005 | |
| Chemical compound, drug | Dasatinib; dasa | Selleck Chemicals | Cat.# S1021 | |
| Chemical compound, drug | Danusertib; danu | Selleck Chemicals | Cat.# S1107 | |
| Chemical compound, drug | Tozasertib; toza | Selleck Chemicals | Cat.# S1048 | |
| Chemical compound, drug | Staurosporine; staur | Selleck Chemicals | Cat.# S1421 | |
| Chemical compound, drug | PD180970 | Sigma-Aldrich | Cat.# PZ0142 | |
| Chemical compound, drug | PD166326 | Sigma-Aldrich | Cat.# PZ0366 | |
| Chemical compound, drug | Rebastinib; reba | Selleck Chemicals | Cat.# S2634 | |
| Chemical compound, drug | Ponatinib; pona | Selleck Chemicals | Cat.# S1490 | |
| Chemical compound, drug | Nilotinib; nilo | Selleck Chemicals | Cat.# S1033 | |
| Chemical compound, drug | Bafetinib; bafe | Selleck Chemicals | Cat.# S1369 | |
| Chemical compound, drug | Imatinib; ima | Selleck Chemicals | Cat.# S2475 | |

*Continued on next page*

*Continued*

| Reagent type (species) or resource | Designation | Source or reference | Identifiers | Additional information |
|---|---|---|---|---|
| Software, algorithm | NMRPipe | *Delaglio et al., 1995* DOI:10.1007/bf00197809 | | |
| Software, algorithm | SPARKY | *Lee et al., 2015* DOI:10.1093/bioinformatics/btu830 | | |
| Software, algorithm | NumPy | *Harris et al., 2020* DOI:10.1038/s41586-020-2649-2 | | |

## Abl constructs, expression, and purification

To generate the various Abl αI-helix mutant constructs (Abl$^{83-513}$, Abl$^{83-514}$, Abl$^{83-515}$, Abl$^{83-516}$, Abl$^{83-517}$, Abl$^{83-518}$, Abl$^{83-519}$, and Abl$^{83-534,E528K}$, Abl 1b numbering), stop (TAG) or lysine (AAG) codons were introduced at the respective positions by site-directed QuikChange mutagenesis into the plasmid containing human Abl$^{83-534}$, which was described earlier (*Skora et al., 2013*). For the initial expression and solubility tests, all Abl constructs were co-expressed with lambda phosphatase (LPP; to obtain dephosphorylated Abl) as described earlier (*Sonti et al., 2018*) in 25 mL LB medium. After harvesting, cells were resuspended in 1 mL lysis buffer, sonicated, centrifuged and the supernatant collected for western blot analysis with a conjugated poly-histidine antibody (Sigma Aldrich, Cat.No. A7058).

All Abl constructs used for NMR experiments were co-expressed with LPP in $^{15}$N-labeled M9 minimal medium as described (*Sonti et al., 2018*). The cell lysate was cleared by centrifugation at 30,000 × *g*, loaded onto a His-Trap column (GE Healthcare), and eluted by applying a linear imidazole gradient from 20 mM to 200 mM over 30 column volumes. Abl$^{83-534}$ was then purified using an ion exchange Q-sepharose HP (GE Healthcare) column equilibrated with 20 mM Tris-HCl, 20 mM NaCl, 2 mM TCEP, 5% glycerol, pH 8.0 (ion exchange buffer). The truncated Abl mutants did not bind to either ion exchange Q- or S-sepharose HP (GE Healthcare) columns in ion exchange buffer. However, using the Q-sepharose HP column helped to remove impurities and the respective Abl constructs were collected in the flow through. In vitro LPP treatment was then applied as described (*Sonti et al., 2018*) in cases where the Abl construct was not fully dephosphorylated as indicated by the elution profile and confirmed by electron-spray ionization (ESI) time-of-flight (TOF) mass spectrometry (MS). All Abl constructs were then further purified by size exclusion chromatography as described (*Sonti et al., 2018*). Final samples were validated by ESI-TOF MS for correct protein mass and absence of contaminations by other proteins.

## NMR spectroscopy

As in previous studies (*Skora et al., 2013*; *Sonti et al., 2018*), the isotope-labeled SH3-SH2-KD Abl constructs were concentrated to 25–100 μM in 20 mM Tris·HCl, 100 mM NaCl, 2 mM EDTA, 2 mM TCEP, 0.02% NaN$_3$, 95% H$_2$O/5% D$_2$O, pH 8.0. Ligands pre-dissolved in DMSO (stock concentration 50 mM) were added at a molar ratio of 3:1 (ligand:protein). All NMR experiments were performed at 303 K on a Bruker AVANCE 900-MHz spectrometer equipped with a TCI triple resonance cryoprobe. $^1$H-$^{15}$N TROSY experiments were recorded with 224 ($^{15}$N)×1024 ($^1$H) complex points and acquisition times of ~40 ms in both dimensions. All NMR data were processed with the NMRpipe software package (*Delaglio et al., 1995*). Spectra were displayed and analyzed with the program SPARKY (*Lee et al., 2015*). The principal component analysis (PCA) of chemical shift variations was carried out using NumPy (*Harris et al., 2020*).

## PDB structure analysis

PDB structures were displayed using the PyMOL Molecular Graphics System (Schrödinger, LLC).

## Activity assays

A biotinylated form of the optimized Abl substrate Abltide (biotin-GGEAIYAAPFKK, obtained from ProteoGenix, France) was incubated in concentrations ranging from 3.125 μM to 200 μM together with 0.1 ng/μl of the respective Abl construct, 100 μM ATP and 5 μCi γ-$^{32}$P-ATP in 20 μl kinase assay buffer

(20 mM Tris-HCl, 5 mM $MgCl_2$, 1 mM DTT, 10 µM bovine serum albumin, pH 7.5) for 12 min at room temperature. The reaction was terminated by adding 10 µl 7.5 M guanidine hydrochloride. Eight µl of the terminated reaction mixture were then applied to a streptavidin biotin capture membrane (SAM2, Promega, Cat.# V2861) and further treated following the manufacturer's recommendations. After a short (~1 min) drying phase, the membrane was washed consecutively 30 s with 2 M NaCl, 3×2 min with 2 M NaCl, 4×2 min with 2 M NaCl +1% $H_3PO_4$, 2×30 s with deionized water, and finally rinsed with ethanol.

The amount of $^{32}P$ incorporated into the Abl substrate was quantified using a liquid scintillation counter (PerkinElmer, model: Tri-Carb4910TR). The Michaelis-Menten parameters and their errors were obtained by Monte Carlo fitting with NumPy (*Harris et al., 2020*) using the sample mean and standard error of the mean of the measured data points.

## Acknowledgements

Drs. Oliver Hantschel, Wolfgang Jahnke, Lukasz Skora, and Layara Akemi Abiko are gratefully acknowledged for helpful discussions. This research has been supported by the Schweizerischer Nationalfonds zur Förderung der Wissenschaftlichen Forschung (grant nos. 31-149927, 31-173089, and 31-201270) and the Krebsliga Schweiz (grant no. KFS-3603-022015).

## Additional information

### Funding

| Funder | Grant reference number | Author |
| --- | --- | --- |
| Schweizerischer Nationalfonds zur Förderung der Wissenschaftlichen Forschung | 31-149927 | Stephan Grzesiek |
| Schweizerischer Nationalfonds zur Förderung der Wissenschaftlichen Forschung | 31-173089 | Stephan Grzesiek |
| Schweizerischer Nationalfonds zur Förderung der Wissenschaftlichen Forschung | 31-201270 | Stephan Grzesiek |
| Krebsliga Schweiz | KFS-3603-022015 | Stephan Grzesiek |

The funders had no role in study design, data collection and interpretation, or the decision to submit the work for publication.

### Author contributions

Johannes Paladini, Conceptualization, Data curation, Formal analysis, Supervision, Investigation, Visualization, Methodology, Writing – original draft, Writing – review and editing; Annalena Maier, Formal analysis, Investigation, Visualization, Writing – original draft; Judith Maria Habazettl, Data curation, Formal analysis, Investigation, Methodology, Writing – review and editing; Ines Hertel, Resources; Rajesh Sonti, Conceptualization, Data curation, Investigation, Methodology; Stephan Grzesiek, Conceptualization, Software, Formal analysis, Supervision, Funding acquisition, Visualization, Writing – original draft, Project administration, Writing – review and editing

### Author ORCIDs

Johannes Paladini ⓘ http://orcid.org/0000-0001-7920-2219
Judith Maria Habazettl ⓘ http://orcid.org/0000-0002-7976-768X
Stephan Grzesiek ⓘ https://orcid.org/0000-0003-1998-4225

Reviewer #1 (Public Review): https://doi.org/10.7554/eLife.92324.3.sa1
Reviewer #2 (Public Review): https://doi.org/10.7554/eLife.92324.3.sa2
Author response https://doi.org/10.7554/eLife.92324.3.sa3

## Additional files

### Supplementary files
• MDAR checklist

### Data availability

NMR time domain data, processing scripts, and peak lists for the various Abl αI-helix mutant constructs in apo form and inhibitor complexes have been deposited in the Biological Magnetic Resonance Data Bank (BMRB) under accession codes 52273, 52293, 52296, 52297, 52298, 52300, 52301, 52302, 52305, 52306, 52307, 52310, 52339.

The following datasets were generated:

| Author(s) | Year | Dataset title | Dataset URL | Database and Identifier |
|---|---|---|---|---|
| Paladini J, Maier A, Habazettl J, Hertel I, Sonti R, Grzesiek S | 2024 | Abl 1b isoform wild type SH3-SH2-KD (aa 83-534) apo | https://bmrb.io/data_library/summary/index.php?bmrbId=52273 | Biological Magnetic Resonance Data Bank, 52273 |
| Paladini J, Maier A, Habazettl J, Hertel I, Sonti R, Grzesiek S | 2024 | Abl 1b isoform wild type SH3-SH2-KD (aa 83-534) in complex with asciminib | https://bmrb.io/data_library/summary/index.php?bmrbId=52293 | Biological Magnetic Resonance Data Bank, 52293 |
| Paladini J, Maier A, Habazettl J, Hertel I, Sonti R, Grzesiek S | 2024 | Abl 1b isoform wild type SH3-SH2-KD (aa 83-519) apo | https://bmrb.io/data_library/summary/index.php?bmrbId=52296 | Biological Magnetic Resonance Data Bank, 52296 |
| Paladini J, Maier A, Habazettl J, Hertel I, Sonti R, Grzesiek S | 2024 | Abl 1b isoform wild type SH3-SH2-KD (aa 83-519) in complex with imatinib | https://bmrb.io/data_library/summary/index.php?bmrbId=52297 | Biological Magnetic Resonance Data Bank, 52297 |
| Paladini J, Maier A, Habazettl J, Hertel I, Sonti R, Grzesiek S | 2024 | Abl 1b isoform wild type SH3-SH2-KD (aa 83-517) apo | https://bmrb.io/data_library/summary/index.php?bmrbId=52298 | Biological Magnetic Resonance Data Bank, 52298 |
| Paladini J, Maier A, Habazettl J, Hertel I, Sonti R, Grzesiek S | 2024 | Abl 1b isoform wild type SH3-SH2-KD (aa 83-517) in complex with imatinib | https://bmrb.io/data_library/summary/index.php?bmrbId=52300 | Biological Magnetic Resonance Data Bank, 52300 |
| Paladini J, Maier A, Habazettl J, Hertel I, Sonti R, Grzesiek S | 2024 | Abl 1b isoform wild type SH3-SH2-KD (aa 83-515) apo | https://bmrb.io/data_library/summary/index.php?bmrbId=52301 | Biological Magnetic Resonance Data Bank, 52301 |
| Paladini J, Maier A, Habazettl J, Hertel I, Sonti R, Grzesiek S | 2024 | Abl 1b isoform wild type SH3-SH2-KD (aa 83-515) in complex with imatinib | https://bmrb.io/data_library/summary/index.php?bmrbId=52302 | Biological Magnetic Resonance Data Bank, 52302 |
| Paladini J, Maier A, Habazettl J, Hertel I, Sonti R, Grzesiek S | 2024 | Abl 1b isoform E528K SH3-SH2-KD (aa 83-534) apo | https://bmrb.io/data_library/summary/index.php?bmrbId=52305 | Biological Magnetic Resonance Data Bank, 52305 |

*Continued*

| Author(s) | Year | Dataset title | Dataset URL | Database and Identifier |
|---|---|---|---|---|
| Paladini J, Maier A, Habazettl J, Hertel I, Sonti R, Grzesiek S | 2024 | Abl 1b isoform E528K SH3-SH2-KD (aa 83-534) in complex with imatinib | https://bmrb.io/data_library/summary/index.php?bmrbId=52306 | Biological Magnetic Resonance Data Bank, 52306 |
| Paladini J, Maier A, Habazettl J, Hertel I, Sonti R, Grzesiek S | 2024 | Abl 1b isoform wild type SH3-SH2-KD (aa 83-534) in complex with GNF-5 | https://bmrb.io/data_library/summary/index.php?bmrbId=52307 | Biological Magnetic Resonance Data Bank, 52307 |
| Paladini J, Maier A, Habazettl J, Hertel I, Sonti R, Grzesiek S | 2024 | Abl 1b isoform wild type SH3-SH2-KD (aa 83-534) in complex with PD166326 | https://bmrb.io/data_library/summary/index.php?bmrbId=52310 | Biological Magnetic Resonance Data Bank, 52310 |
| Paladini J, Maier A, Habazettl J, Hertel I, Sonti R, Grzesiek S | 2024 | Abl 1b isoform wild type SH3-SH2-KD (aa 83-534) in complex with imatinib | https://bmrb.io/data_library/summary/index.php?bmrbId=52339 | Biological Magnetic Resonance Data Bank, 52339 |

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
