## [Editor Report · eLife assessment]

This manuscript describes an **important** NMR investigation of allosteric interactions within Abl kinase. The authors identify helix I as a major element that couples the Abl active site with the myristate-binding pocket. The **convincing** findings have implications for understanding Abl kinase activation and how to target Abl kinase in diseases.

---

## [Referee Report · Reviewer #1 (Public Review)]

Summary:

The authors identify a mechanical model of activation of Abelson kinase involving the modification of stability of an alpha helix by mutations and different classes of inhibitors. They use NMR chemical shifts of mutant sequences of the alpha helix in a model of Abelson kinase including the regulatory and kinase domains.

Strengths:

The mechanism of inhibition of this important drug target is highly complex involving multiple domains' interactions, While crystal structures can establish end states well, the details of more dynamic interactions among the components can be assessed by NMR studies, The authors previously established {Sonti, 2018, PMID29319304} that different inhibitors and assembled states result from changes of stabilisation of the assembly involving the kinase and the SH3 domain. This is extended here to illuminate the role of the kinase C terminal alpha helic I' to the domains' interface, expanding the previous identification of this area of the protein as key to agonist/antagonist action at the allosteric myristlylation binding site.

---

## [Referee Report · Reviewer #2 (Public Review)]

In this paper, Paladini and colleagues investigate the concerted motions within the Abl kinase that control its conformational transition between the active (disassembled) and inactive (assembled state). This work follows their previously published findings that binding of the type II inhibitor, imatinib to the active site of Abl, leads to kinase core disassembly via the force imposed by the P-loop and other regions of the N-lobe on the SH3 domain. Interestingly, imatinib-induced disassembly is prevented when an allosteric inhibitor, asciminib, binds to the myristate-binding pocket. Key to asciminib and myristate binding are motions of helix I, located in the C-lobe, and thus, helix I is hypothesized to be the sensor of the imatinib-induced changes. Specifically, bending of helix I upon engagement of myristate or asciminib was postulated to be important for re-assembly of the autoinhibited Abl core, and thus, reducing the "force" with which kinase N-lobe pushes against the SH2 domain upon binding imatinib.

The authors use NMR to measure conformational transitions in the several 15N-labeled Abl kinase constructs that display different degrees of helix I truncations. This analysis is slightly limited by the instability of the constructs that carry truncations beyond the helix I "bend". Nevertheless, it is sufficient to establish that truncation of helix I that removes its fragment, which is in contact with myristate or asciminib ligands, results in loss of the ability of helix I to impose "force" on the SH2 domain that results in kinase core disassembly, even in the presence of imatinib binding. In the absence of this force, the allosteric coupling between the helix I/SH2 and KD/SH3 interfaces is compromised. Principle component analysis is used to analyze the NMR data, and it is very clear and convincing.

A compelling evidence in support of the proposed allosteric mechanism comes from the analysis of the E528K disease mutation, identified in the Abl1 malformation syndrome. The authors show that this mutant, poised to break a salt bridge formed between E528 in the C-terminal portion of helix I and R479 on the kinase domain, increases helix I outward motions resulting in core disassembly and higher Abl kinase activity. Together, these results reinforce that helix I motions are central to the mechanism of kinase activation via core disassembly.

---

## [Author Response]

The following is the authors’ response to the original reviews.

**Reviewer #1 (Recommendations For The Authors):**
This is significant work, and you should certainly make the best case you can on the weaknesses discussed.

We thank reviewer for this positive comment on the significance of our work. The referee indicates as weaknesses (i) that the force involving the bent or straight αI-helix is not readily apparent, (ii) the residue types were not varied in the helix mutations, and (iii) that the chemical shift perturbations are indirect observations.

We think we have tried to address a large part of these questions by being very careful in our analysis and by the discussion in the manuscript. The following remarks may help to clarify this further:

(i) The force emanating from the helix is e.g. visualized in the PC2 loadings in Figure 6E of the PCA carried on all observed SH3-SH2-KD resonances for all apo forms of the helix mutants. The SH2 residues identified by these loadings are in direct vicinity to the αI-helix. The respective PC2 scores correlate to 98% with the vmax of the catalytic reaction and to 94 % with the PC1 scores found for imatinib-induced opening. Importantly, the structure of the KD with the straight αI-helix indicates that mostly residues F516, Q517, S520, and I521 would clash with the SH2 domain in a closed core (Figure 6F). Thus, the expected clashes are in direct vicinity of the SH2 residues identified by the PC2 loadings as correlated to vmax and imatinib-induced opening. These data are completely orthogonal and show that most of the force is coming from residues F516, Q517, S520, and I521 in the αI-αI’ turn.

(ii) We agree that we mainly used truncations of the αI-helix to study its involvement in activation. Point (i) makes it clear that a larger part of the αI-helix effects is caused by steric clashes of the residues in the αI-αI’ turn. In the latter region, we don’t expect strong amino acid type-specific effects besides excluded volume. Due to expression problems, we could not vary the helix length between residues 519 and 534. However, in this region we introduced the amino acid type mutation E528K. The latter showed a clear specific effect. Further amino acid type-specific effects may be possible in this region. However, we expect that the identified electrostatic E528-R479 interaction is one of the most important interactions in this region.

(iii) We agree that chemical shift changes of individual resonances are often hard to interpret. However, we want to stress that our conclusions are all drawn from principal component analyses, which in all cases had as input well over 100 if not over 200 1H-15H resonances. The first two principal components of these analyses are robust averages over many residues, which reveal general correlated structural trends.

We assume that chemical shift deposition etc will be pursued.

We are currently depositing a larger collection of our Abl data to the “Biological Magnetic Resonance Data Bank (BMRB)”, which includes the NMR chemical shift data of the present work. A ‘collection’ will be a new feature of the BMRB, and we are in discussion with their staff. We will provide the accession codes as soon as possible (probably within the next month) to be included into the final version of the manuscript. We have amended the Data Availability Section accordingly.

**Reviewer #2 (Recommendations For The Authors):**
1. The overall discussion of the implications of the described allostery on kinase activation is provided through lenses of imatinib binding, which is used as an experimental trigger to disassemble the autoinhibited core. Can the authors elaborate in the Discussion on what event would play this role in the kinase catalytic cycle, communicating to helix I? Would dissociation of the myristate from the active site be hypothesized to be the first step in kinase activation? While I understand that certainty may be challenging to attain, it would be good to introduce some ideas into the Discussion.

We appreciate the reviewer’s suggestions for the discussion and added the following text to the Conclusion section:

"We have used here imatinib binding to the ATP-pocket as an experimental tool to disassemble the Abl regulatory core. Our previous analysis (Sonti et al., 2018) of the high-resolution Abl transition-state structure (Levinson et al., 2006) indicated that due to the extremely tight packing of the catalytic pocket, binding and release of the ATP and tyrosine peptide substrates is only possible if the P-loop and thereby the N-lobe move towards the SH3 domain by about 1–2 Å. This motion is of similar size and direction as the motion of the N-lobe observed in complexes with imatinib and other type II inhibitors (Sonti et al., 2018). From this we concluded that substrate binding opens the Abl core in a similar way as imatinib. The present NMR and activity data now clearly establish the essential role of the αI-helix both in the imatinib- and substrate-induced opening of the core, thereby further corroborating the similarity of both disassembly processes.

Notably, the used regulatory core construct Abl83-534 lacks the myristoylated N-cap. Although we have previously demonstrated that the latter construct is predominantly assembled (Skora et al., 2013), the addition of the myristoyl moiety is expected to further stabilize the assembled conformation in a similar way as asciminib.

Considering this mechanism, dissociation of myristoyl from the native Abl 1b core may be a first step during activation. However, it should be kept in mind that the Abl 1a isoform lacks the N-terminal myristoylation, and it is presently unclear whether other moieties bind to the myristoyl pocket of Abl 1a during cellular processes."

1. Can the authors comment more on the differentiation between assembled conformations induced by type I inhibitor binding vs apo forms (or AMP-PNP and allosteric inhibitor) reported in Figure 3B? The differences are clearly identified by PCA but not sufficiently discussed.

As indicated in the text, we think two structural effects are intermingled within PC2. Due to this admixture, it is hard to draw strong conclusions and we don’t want to expand on this too much. We have slightly modified the respective paragraph (p.7) as follows:

"As the affected residues react differently to perturbations by type I inhibitors and truncation of the αI’-helix (Figure 3A, right), we attribute this behavior to two effects intermixed into the PC2 detection: (i) a minor rearrangement of the SH3/KD N-lobe interface caused by filling of the ATP pocket with type I inhibitors, which in contrast to the stronger N-lobe motion induced by type II inhibitors does not yet lead to core disassembly and (ii) a small rearrangement of the SH2/KD C-lobe interface caused by shortening and mutations of the αI-helix."

1. The allosteric connection between active site inhibitor binding and the myristate/allosteric inhibitor binding has been observed in the past and noted before, in papers such as Zhang et al, Nature 2010. While the authors reference this paper, they do not acknowledge its specific findings or engage in a broader discussion of how their conclusions relate to this work.

We have modified the beginning of the Conclusion section:

"The allosteric connection between Abl ATP site and myristate site inhibitor binding has been noted before, albeit specific settings such as construct boundaries and the control of phosphorylation vary in published experiments. Positive and negative binding cooperativity of certain ATP-pocket and allosteric inhibitors has been observed in cellular assays and in vitro (Kim et al., 2023; Zhang et al., 2010). Furthermore, hydrogen exchange mass spectrometry has indicated changes around the unliganded ATP pocket upon binding of the allosteric inhibitor GNF-5 (Zhang et al., 2010). Here, we present a detailed high-resolution explanation of these allosteric effects via a mechanical connection between the kinase domain N- and C-lobes that is mediated by the regulatory SH2 and SH3 domains and involves the αI helix as a crucial element.

Specifically, we have established a firm correlation between the kinase activity of the Abl regulatory core, the imatinib (type II inhibitor)-induced disassembly of the core, which is caused by a force FKD–N,SH3 between the KD N-lobe and the SH3 domain, and a force FαI,SH2 exerted by the αI-helix towards the SH2 domain. The FαI,SH2 force is mainly caused by a clash of the αI-αI’ loop with the SH2 domain. Both the FKD–N,SH3 and FαI,SH2 force act on the KD/SH2SH3 interface and may lead to the disassembly of the core, which is in a delicate equilibrium between assembled and disassembled forms. As disassembly is required for kinase activity, the modulation of both forces constitutes a very sensitive regulation mechanism. Allosteric inhibitors such as asciminib and also myristoyl, the natural allosteric pocket binder, pull the αI-αI’ loop away from the SH2 interface, and thereby reduce the FαI,SH2 force and activity. Notably, all observations described here were obtained under nonphosphorylated conditions, as phosphorylation will lead to additional strong activating effects."

1. Figure 6 could do a better job of providing an illustration of steric clashes.

We have revised Figure 6, panel F, in order to better illustrate the steric clashes, and modified the legend accordingly.

1. There is a typo in line 5 from the top on page 11 (dash missing from "83534" superscript).

Thank you. This was fixed.